# Optimization of Culture Conditions for Oxygen-Tolerant Regulatory [NiFe]-Hydrogenase Production from *Ralstonia eutropha* H16 in *Escherichia coli*

**DOI:** 10.3390/microorganisms9061195

**Published:** 2021-05-31

**Authors:** Qin Fan, Giorgio Caserta, Christian Lorent, Oliver Lenz, Peter Neubauer, Matthias Gimpel

**Affiliations:** 1Institute of Biotechnology, Technische Universität Berlin, Chair of Bioprocess Engineering, Ackerstraße 76, D-13355 Berlin, Germany; qin.fan@campus.tu-berlin.com (Q.F.); peter.neubauer@tu-berlin.de (P.N.); 2Department of Chemistry, Technische Universität Berlin, Straße des 17. Juni 135, D-10623 Berlin, Germany; giorgio.caserta@tu-berlin.de (G.C.); christian.lorent@tu-berlin.de (C.L.); oliver.lenz@tu-berlin.de (O.L.)

**Keywords:** [NiFe]-hydrogenase, *Ralstonia eutropha*, heterologous protein production, cofactor assembly, difficult-to-express protein, *Escherichia coli*

## Abstract

Hydrogenases are abundant metalloenzymes that catalyze the reversible conversion of molecular H_2_ into protons and electrons. Important achievements have been made over the past two decades in the understanding of these highly complex enzymes. However, most hydrogenases have low production yields requiring many efforts and high costs for cultivation limiting their investigation. Heterologous production of these hydrogenases in a robust and genetically tractable expression host is an attractive strategy to make these enzymes more accessible. In the present study, we chose the oxygen-tolerant H_2_-sensing regulatory [NiFe]-hydrogenase (RH) from *Ralstonia eutropha* H16 owing to its relatively simple architecture compared to other [NiFe]-hydrogenases as a model to develop a heterologous hydrogenase production system in *Escherichia coli*. Using screening experiments in 24 deep-well plates with 3 mL working volume, we investigated relevant cultivation parameters, including inducer concentration, expression temperature, and expression time. The RH yield could be increased from 14 mg/L up to >250 mg/L by switching from a batch to an EnPresso B-based fed-batch like cultivation in shake flasks. This yield exceeds the amount of RH purified from the homologous host *R. eutropha* by several 100-fold. Additionally, we report the successful overproduction of the RH single subunits HoxB and HoxC, suitable for biochemical and spectroscopic investigations. Even though both RH and HoxC proteins were isolated in an inactive, cofactor free apo-form, the proposed strategy may powerfully accelerate bioprocess development and structural studies for both basic research and applied studies. These results are discussed in the context of the regulation mechanisms governing the assembly of large and small hydrogenase subunits.

## 1. Introduction

Hydrogenases can be found in many bacteria and archaea as well as in a few unicellular eukaryotes [1]. According to the metal composition of the active site, they are classified as [Fe]-, [FeFe]-, and [NiFe]-hydrogenases [1]. Among the members of [NiFe]-hydrogenases, O_2_-tolerant enzymes are particularly attractive as they remain catalytically active under oxic conditions, unlike all other hydrogenases [2]. This facilitates their biotechnological application in the areas of biohydrogen production, cofactor regeneration, and fuel cells [3,4,5]. All so-far isolated [NiFe]-hydrogenases consist of at least two subunits; a large subunit (LSU) of approximately 60 kDa housing the catalytic center and a small subunit (SSU) of approximately 30 kDa hosting one to three iron-sulfur (Fe–S) clusters [6]. The Fe–S clusters serve as an electron relay transferring reducing power towards or away from the catalytic center [7]. The catalytic center carries the heterobimetallic [NiFe] site, in which a nickel and an iron ion are bridged by two cysteines. Two further cysteines act as terminal ligands to Ni, while one CO and two CN^−^ ligands participate in the coordination of the Fe [8]. The highly complicated [NiFe(CN)_2_CO]-cofactor assembly of hydrogenases requires a dedicated set of at least six maturation Hyp proteins and a specific endopeptidase [2,9].

The β-proteobacterium *Ralstonia eutropha* H16 (also named *Cupriavidus necator*) harbors four different O_2_-tolerant [NiFe]-hydrogenases; the membrane-bound hydrogenase (MBH), the soluble NAD^+^-reducing hydrogenase (SH), the actinobacterial-like hydrogenase (AH), and the regulatory hydrogenase (RH) [10]. Here, we focus on the RH that functions as H_2_ sensor regulating the expression of energy-converting MBH and SH proteins [2]. In addition to the RH heterodimer, consisting of the HoxB and HoxC subunits, the H_2_-sensing two-component regulatory system includes the histidine kinase HoxJ and the response regulator HoxA [11] (Figure 1). Considering the high complexity of the RH system depicted in Figure 1, many in vitro studies, e.g., EPR [12], IR [13,14,15], Mössbauer [16], resonance Raman (RR) [17], and nuclear resonance vibrational (NRVS) [18] spectroscopy, have been carried out on a truncated version of the protein (RH_stop_, Figure 1 right side), which allows for the isolation of the single HoxBC heterodimer [19,20]. Nevertheless, relatively low protein yields of 0.01–0.1 mg of purified enzyme per L of culture have been reported for both homologous (*R. eutropha)* [13,19,20] and heterologous (*E. coli*) [21] RH_stop_ production.

The heterologous production of hydrogenases is a challenging task [22]. However, the relatively simple architecture of the RH compared to other [NiFe]-hydrogenases and its utilization as a model system for the investigation of the catalytic cycle of [NiFe]-hydrogenases [14,23,24], prompted us to develop new strategies for heterologous high-yield productions of RH_stop_ in *E. coli* as a model for bioprocess development of heterologous [NiFe]-hydrogenase production.

Here, we report on determining parameters for improved RH overproduction resulting in the isolation of high amounts of soluble RH_stop_ protein as well as its isolated large subunit HoxC and small subunit HoxB. We used the enzyme-based glucose-release strategy (EnBase^®^) for high-yield RH production under small-scale fed-batch like conditions and, subsequently, for the successful overproduction of the RH single subunits HoxB and HoxC, suitable for biochemical and spectroscopic investigations. Surprisingly, the large and small subunit hydrogenase assembly was observed in the absence of the catalytic [NiFe]-center. These results are discussed in the context of the still unknown maturation mechanism and the regulation factors governing the assembly of large and small hydrogenase subunits.

## 2. Materials and Methods

### 2.1. Bacterial Strains and Growth Conditions

*E. coli* TG1 (Appendix A) [25] was used for cloning. For HoxBC overproduction, we used *E. coli* BL21-Gold (Agilent, Waldbronn, Germany). Unless stated otherwise, *E. coli* cultivations were performed at 30 °C in an incubation shaker at 250 rpm. Terrific broth (TB) [26] was used for batch cultivations. EnPresso B (EnPresso GmbH, Berlin, Germany) was used for fed-batch-like cultivations [27]. For all *E. coli* cultivations, 25 µg/mL chloramphenicol were added.

### 2.2. Plasmid Design and Construction

For the construction of a *hox* gene expression plasmid, we first removed the unique *BstBI* site of plasmid pCTUT7 (Appendix A) [28] by integration of the oligonucleotides MG0034 and MG035 (Appendix A) into BstBI-cut pCTUT7, yielding plasmid pQF1. Subsequently, the *Bacillus subtilis*
*bsrF* transcription terminator [29] was integrated between the unique HindIII and BamHI sites using the annealed oligonucleotides MG0051 and MG0052, yielding plasmid pQF3. To generate a *hoxB* version that encodes a Strep-tag II sequence instead of the C-terminal HoxJ-binding domain, two subsequent PCR reactions were performed, the first on plasmid pCH594 [23] and primer pair MG0038/MG0039. The resulting fragment served as template for a second PCR reaction using the primer pair MG0038/MG0040. The resulting 0.9 kb fragment was digested with XbaI and HindIII and ligated into the corresponding pQF3 vector yielding plasmid pQF4. The primer pair MG0043/MG0044 and plasmid pCH594 as template were used to PCR-amplify a 1.5 kb fragment encoding the C-terminally Strep-tagged HoxC. The fragment was digested with XbaI and HindIII and ligated into pQF3, resulting in plasmid pQF5. For co-production of both RH subunits, an additional plasmid was constructed. To this end, a PCR was performed with plasmid pQF5 as the template and primer pair MG0041/MG0042 to generate a fragment encoding untagged HoxC. The amplicon was digested with BstBI and HindIII, and the resulting fragment was integrated into the likewise cut plasmid pQF4, yielding plasmid pQF8 (encoding HoxB_Strep_C). The correctness of all constructs was verified by sequencing (LGC Genomics, Berlin, Germany).

### 2.3. Deepwell Plate Cultivation

Small-scale expression screenings were carried out in square-bottom 24-deepwell plates (24xDWP; Thomson Instrument Company, Oceanside, CA, USA) with a working volume of 3 mL at 37 ^°^C in an orbital shaker. For optimal gas exchange, plates were covered with the Duetz System sandwich cover and clamp system (Enzyscreen, Er Haarlem, The Netherlands). Cultures were inoculated to an initial OD_600_ of 0.2 from a fresh overnight preculture in an Ultra Yield Flasks™ (UYF; Thomson Instrument Company, Oceanside, CA, USA) shaken at 37 °C and 250 rpm. As OD_600_ reached 0.8–1.0, 3 mL of respective cultures were transferred into the 24xDWP and expression was induced with varying concentrations of IPTG inducer (0, 5, 20, 50, 100, 500 µM, and 1 mM). Cells were harvested 5 h after induction by centrifugation for 10 min at 16,000× *g* at 4 °C and subjected to SDS-PAGE and Western blot analysis immediately. When using EnPresso B culture medium, for each well, boosting nutrients and a second dose of 1.5 U/L Reagent A were added at the induction time according to the manufacturer’s instructions. Cells were cultivated for 24 h after induction in the shaker (Infor HT, 25 mm offset, Switzerland) at 30 °C, 250 rpm. Then, 0.5 mL samples were taken 6 and 24 h after induction by centrifugation for 10 min at 16,000× *g* at 4 °C. The cell pellets were stored at −20 °C until further use. Optical densities (OD_600_) of the cultures were determined after manually diluting 50–200 µL of the culture with 0.9% (*v*/*v*) NaCl solution and measuring the absorption at 600 nm using a Ultrospec 2100 pro spectrophotometer (Amersham Bioscience, Switzerland).

### 2.4. Shake-Flask Cultivation

Expression in the shake flask scale was carried out in baffled UYFs with TB or EnPresso B medium (filled to 20% of their maximum volume). All UY-flasks were sealed with adhesive air-permeable membranes (AirOTop; Thomson Instrument, Oceanside, CA, USA), and incubated with orbital shaking at 30 °C, 250 rpm. EnPresso B medium was used for fed-batch-like cultivation according to the manufacturer’s recommended procedure (EnPresso GmbH, Berlin, Germany). Cultivations were performed in 50 mL or 500 mL EnPresso B medium in 250 mL or 2.5 L UY-flasks. A dose of 1.5 U/L glucose-releasing biocatalyst Reagent A was added to the medium immediately followed by inoculation to an OD_600_ of 0.2 from a log-phase LB culture. After overnight cultivation (15–18 h), target gene expression was induced by IPTG addition. If required, a booster and a second dose of 4.5 U/L Reagent A were added at the induction point. After induction, cells were cultivated for another 24 h and were then harvested by centrifugation for 10 min with 8000× *g* at 4 °C. Similarly, batch cultivations in 500–1000 mL complex TB medium were performed using 2.5 L UYF. Here, target gene expression was induced by addition of IPTG at OD_600_ of approximately 0.8. After induction, cells were cultivated for another 18–24 h and harvested as stated above. In all cases, 0.1 mL/L Antifoam 204 (0.01% *v*/*v*; Sigma-Aldrich, Steinheim, Germany) was added to prevent foam formation during UYF cultivations and OD_600_ was measured as described above.

### 2.5. Purification of Strep-Tagged Hox Proteins

Purification of the Strep-tagged proteins using high-capacity Strep-Tactin Superflow (IBA, Göttingen, Germany) was performed according to the manufacturer’s instructions. Briefly, cell pellets were resuspended in washing buffer A (100 mM Tris-HCl, pH 8.0, 150 mM NaCl) (4 mL per g wet cell weight) supplemented with 1 mg/mL lysozyme and 1 mM PMSF and subsequently disrupted by sonication (8–10 min for each 20 mL of suspension; 30 s on/off, sonotrode with 7 mm diameter, 60% amplitude). Crude extracts were centrifuged for 30 min at 16,000× *g* at 4 °C and the soluble fraction was loaded onto a Gravity flow Strep-Tactin column. The column was washed with five bed volumes of washing buffer A, and the proteins were eluted with six bed volumes of buffer A containing 2.5 mM D-desthiobiotin. If required, eluates were concentrated by ultra-filtration (14,000× *g*, 4 °C) using Amicon Ultra Ultracel 30 kDa cut-off concentrators (Merk Millipore, Darmstadt). The final concentrate was flash-frozen in liquid nitrogen and stored at −80 °C. SDS-PAGE and subsequent quantification of the bands with ImageJ was used for determination of protein concentrations. A solution of bovine serum albumin served as the standard. All gels were visualized by staining with modified colloidal Coomassie blue G250 as reported elsewhere [30].

### 2.6. Western Blot Analysis

Proteins were separated in 12% SDS-PAGE gels. Semi-dry Western blotting was performed as described previously [31]. To prevent nonspecific binding of antibodies, the PVDF membrane was blocked with 2% *v/v* BSA in PBST buffer (137 mM NaCl, 2.7 mM KCl, 10.2 mM Na_2_HPO_4_, pH 7.4, 0.2% Tween 20). Strep-tagged proteins were immunologically detected using specific antibodies against the Strep tag (Iba GmbH, Germany) in a 1:2500 dilution and alkaline phosphatase-conjugated goat anti-mouse IgG (Sigma-Aldrich, Germany) in a 1:8000 dilution. Visualization of the membranes was achieved via the AP catalyzed reaction of the chromogenic BCIP/NBT substrates [80 µL BCIP (20 mg/mL BCIP in 100% DMF) plus 60 µL NBT (50 mg/mL NBT in 70% DMF) in 10 mL reaction buffer]. Blots were digitized with a Scan-Prisa 640U scanner and quantified with ImageJ.

### 2.7. Spectroscopic Analysis and Activity Assays of RH

H_2_-mediated reduction of methylene blue was measured spectrophotometrically as previously described [32]. The reaction was performed in a 3-mL glass cuvette (1 cm path length) with a rubber septum at 30 °C. The methylene blue was added to the cuvette containing the reaction buffer (50 mM K_2_HPO_4_/KH_2_PO_4_, pH 7.0) to a final concentration of 200 µM. After saturating the reaction mixture with H_2_, the protein samples (50–300 µM) were injected. The specific H_2_-dependent reduction of methylene blue was monitored at 570 nm using a Cary 50 UV-vis spectrophotometer (Varian, Agilent) and a molar extinction coefficient of 13.1/(mM cm). Hydrogenase activity is expressed in Units (U, corresponding to 1 µmol of H_2_ oxidized per min) per mg of protein. Each activity measurement lasted 20–30 min.

UV-vis absorption spectra were recorded on a Cary 300 UV-vis spectrophotometer (Varian, Agilent). The spectra were recorded between 250 nm and 800 nm using buffer A as reference. UV-vis spectra of HoxC were recorded using 200 µM and 20 µM protein solutions, respectively. Buffer A was used to adjust the protein concentrations.

EPR of RH was performed at a concentration of 0.2 mM. The measurements were carried out as previously described [13,14].

HoxC and RH proteins (0.2–0.4 mM) were transferred into a gas-tight, temperature-controlled (10 °C), transmission cell (volume ca. 7–10 µL) equipped with two sandwiched CaF_2_ windows, separated by a Teflon spacer (50 µm path length). Spectra with a 2/cm resolution were recorded on a Tensor 27 Fourier-transform spectrometer from Bruker equipped with a liquid nitrogen cooled mercury cadmium telluride (MCT) detector as previously described [14]. A buffer spectrum was used as a reference to calculate the corresponding absorbance spectra. The Bruker OPUS software 7.5 package (Bruker Optics) was used for data acquisition and evaluation.

## 3. Results

### 3.1. Construction of Expression Vectors for Heterologous Hydrogenase Production in E. coli

Because of their complex structure and the sophisticated biosynthesis process, development of a bioprocess for heterologous production of [NiFe]-hydrogenase is a quite challenging task. For construction of the required expression plasmids, the pBR322 derivative pCTUT7 [28] was chosen as the basis. The plasmid carries the strong P_lac_ promoter derivative P_lac-CTU_ [28] that allows IPTG-inducible hydrogenase gene expression. Plasmid pQF8 (Appendix A) harbors the RH structural genes *hoxB* and *hoxC* arranged as an operon under control of the P_lac-CTU_ promoter. A Strep-tag II replaces the C-terminal domain of HoxB responsible for interaction with the histidine kinase HoxJ [33] and allows facile purification of the RH_Stop_ by affinity chromatography. The HoxC protein remained untagged. In addition, we constructed plasmids pQF4 and pQF5 (Appendix A), which encoded the single structural hydrogenase subunits HoxB and HoxC, respectively, each with a C-terminal Strep tag II sequence.

First expression experiments using *E. coli* strains TQF4B, TQF5C, or TQF8RH (*E. coli* TG1 carrying plasmids pQF4, pQF5, or pQF8, respectively) were performed at 37 °C in shake flasks containing TB complex medium. In all cases, the *hoxB* and *hoxC* genes were overexpressed and the corresponding proteins could be purified by affinity chromatography (Appendix A), demonstrating the functionality of these plasmids. The bands corresponded to the calculated theoretical molecular masses of 32 kDa, 54 kDa, and 52 kDa for HoxB_Strep_, HoxC_Strep,_ and untagged HoxC, respectively. In case of HoxB, however, the yield was comparatively low and a second band directly below HoxB was detected in the preparation from TQF4B, which might indicate proteolytic degradation of HoxB as recently shown (Appendix A) [34]. This was confirmed by immunoblotting, as both bands reacted with the anti-Strep-tag antibody. Nonetheless, the second HoxB band was not observed in RH samples purified from stain TQF8RH, possibly due to the presence of HoxC that might help to protect HoxB from degradation. Moreover, in all samples of purified Hox proteins, a prominent additional band with a size of approximately 60 kDa was visible.

### 3.2. Screening of Inducer Concentrations for HoxBC Overproduction in E. coli

The low HoxB yields and the presence of contaminating proteins in purifications from strain *E. coli* TG1 prompted us to evaluate the performance of *E. coli* strain BL21-Gold, which is frequently used for heterologous protein production. We carried out batch cultivations with 3 mL TB medium in square bottom deep-well plates and induced *hox* gene expression in strains BQF4B, BQF5C and BQF8RH by the addition of different IPTG concentrations (see Materials and Methods). As expected, increasing IPTG concentrations resulted in higher total Hox protein production (Figure 2). However, in all three cases, 50 µM IPTG were sufficient to produce more than 80% of the maximum yield obtained with 1 mM IPTG (Figure 2). Moreover, the additional HoxB band observed when using *E. coli* TG1 for overproduction (see above and Appendix A) was not detectable in HoxB preparations from strains BQF4B or BQF8RH (Figure 2), indicating higher stability of the recombinant protein in the B-strains, which might be attributed to the absence of the OmpT and Lon proteases [35,36]. Consequently, this condition was used for the following scale-up experiments in Ultra Yield flasks (UYF).

### 3.3. Low Inducer Concentrations and Low Temperatures Support Production of Soluble RH

Based on the promising screening results, we moved on to process upscaling. To this end, we performed experiments with strain BQF8RH cultivated in a TB medium in UYFs. Expression of the RH genes was induced by 1 mM IPTG as this resulted in the highest RH production in the previous screening experiments (see above). SDS-PAGE analysis of the RH subunits purified by affinity chromatography revealed that the yield of both HoxB and HoxC was relatively low compared to that of cultures induced with 50 µM IPTG (Appendix A). This low yield might be attributed to a loss of soluble protein due to the formation of inclusion bodies under the conditions used, which in turn might be due to an unsuitable cultivation temperature [37] or an inducer concentration that was too high [38]. To investigate the temperature effect, we reduced the cultivation temperature from 37 °C to 30 °C. In fact, we observed about 3.5-fold higher protein levels (Figure 3A) upon cultivation at 30 °C. Even growth temperatures below 30 °C could be applied to facilitate soluble protein production [39,40]. However, a further decrease of the expression temperature from 30 °C to 20 °C did not increase the yield of soluble RH protein (Appendix A). While the growth time at 20 °C reaching the same final OD_600_ increased by almost 1.5-fold compared to 30 °C, the space-time yield of purified RH decreased by almost half (Appendix A).

Next, we investigated the impact of the inducer concentration on the yield of soluble recombinant RH. To this end, we induced protein production with either 1 mM or 50 µM IPTG. Interestingly, the lower inducer concentration improved the yield of soluble RH about 2-fold at both 30 °C and 37 °C (Figure 3A). Similar amounts of HoxB_Strep_ were produced in whole cells irrespective of the temperature and the inducer concentration (Figure 3B). However, significantly higher yields were obtained when RH_Stop_ was purified from cells grown at a lower temperature and a lower inducer concentration. Similarly, the two single RH subunits HoxB and HoxC could be produced in soluble form under these conditions using strains BQF4B and BQF5C, respectively (Appendix A).

Taken together, our results indicate that the RH as well as their single subunits can be recombinantly produced in *E. coli* and the purification of soluble RH succeeds better at lower growth temperatures and lower inducer concentrations, which seem to lower the formation of inclusion bodies.

### 3.4. Optimization of Expression Time

The determination of the optimal expression time is also important to increase the overall productivity of the process. To address this question, the Hox protein production levels were investigated in TB batch cultures of strains BQF4B, BQF5C, and BQF8RH, respectively. Hox protein production was induced by addition of 50 µM IPTG at OD_600_ of approximately 0.8, and the amount of produced Hox protein at different time points was analyzed by Western blotting (Figure 4). As expected, the Hox protein level increased during the first 10 h after induction for all three strains. The highest amount of total Hox protein was reached after 10 h of induction (Figure 4). No further increase in protein level was detected in TB batch cultivations upon prolonged induction for 23 h at the time when the cell growth also stopped (Figure 4). Consequently, the way to obtain high cell densities by prolonged cell growth, rather than prolonged induction time, is to increase HoxBC production, which is the subsequent aim of this study.

### 3.5. Fed-Batch Like Cultivation for Heterologous RH Production in E. coli

Growth of *E. coli* in conventional complex media does not result in high cell densities [41,42,43]. Therefore, we applied the enzyme-based glucose delivery EnBase technology for growing our recombinant strains [27]. In this cultivation process, the continuous and automated gradual release of glucose into the medium, catalyzed by a glucoamylase, and the presence of pH-stabilizing substituents, result in high cell densities (OD_600_ = 20–50), which generally leads to higher yields of recombinant protein in small-scale shake flasks [44,45,46]. In addition, this technology is easy to scale-up to real, high-cell-density, fed-batch cultivations.

To improve the RH production performance of strain BQF8RH, we used the EnPresso B growth system [44,45] with or without addition of booster tablet and a second dose of glucoamylase (EnPresso GmbH, Germany) prior to induction. After growth for 18 h, gene expression was induced trough IPTG addition in concentrations ranging from 0–1 mM. Equal amounts of cells were harvested after 24 h of induction, and the individual HoxB levels were determined by Western blotting (Figure 5A). In both boosted and non-boosted cultures, respectively, increasing inducer concentrations resulted in increasing HoxB levels (Figure 5B), as already observed in TB complex medium (Figure 2). As illustrated in Figure 5B, both the boosted and the non-boosted cultures show comparable maximum HoxB yields at the highest inducer concentration. Interestingly, about 90% of the maximum HoxB yield was already attained with 50 µM IPTG in the boosted Enpresso B culture, whereas only about 45% of the maximum HoxB yield was reached in the non-boosted culture upon induction with 50 µM IPTG, indicating a positive effect of the booster on RH production at lower inducer concentration. In contrast to the batch cultivation, less leaky expression was observed in the EnPresso cultures without induction (compare Figure 2 and Figure 5). Not surprisingly, due to the glucose-limited growth, the non-boosted cultures reached final OD_600_s that were approximately 1.5-fold lower compared to the boosted cultures (Figure 5C). Consequently, the volumetric RH yield was about 2-fold higher in the boosted compared to non-boosted cultures (Figure 5D). A similar result of an about 2-fold increase in cell density was observed in 50-mL UYF cultures with and without booster (Figure 5D). In contrast to non-boosted cultures, the soluble RH yield per cell was slightly increased in boosted cultures, whereas approximately 2-fold higher volumetric RH yield was obtained largely due to the overall increase in the cell density (Figure 5E,F). Our results indicate that the fed-batch like growth system is suitable for inducer-dependent RH production in *E. coli* through an automated, enzyme-based glucose-release into the cultures.

### 3.6. Upscalable RH Production in Shake Flask in Batch and Fed-Batch Cultivations

To investigate the reproducibility of the batch und fed-batch like cultures during scale-up, we performed shake flask cultures in 2.5-L UYF with a broth volume of 500 mL (20% *v*/*v*) under optimized conditions of inducer concentration, temperature, and production strain. At the end, the cells grew slower in 2.5-L UYFs most likely due to lower oxygen transfer coefficient (K_L_a) values (at least 3.5-times lower) than that in 250-mL UYFs [47]. Interestingly, the slower growth positively affected RH production in both media. We were able to achieve RH yields of 4 mg/g cell wet weight in TB and even 6 mg/g in boosted EnPresso medium (Figure 6B). The final titer was increased up to approximately 255 mg/L in EnPresso cultivations, which represents a 2-fold higher yield compared to the 500 mL TB batch culture (Figure 6) and a 2-fold increase compared to the 50 mL EnPresso cultures (Figure 5F). From these results, we conclude that a good scalability of RH production could be achieved over the shake-flask scales (250 mL vs. 2.5 L shake flasks) in terms of the volumetric and specific protein yields. A lower K_La_ value may seem to be advantageous for scale-up and facilitate RH production.

Taken together, switching from a batch to a fed-batch like system with an enzyme-based automated in situ glucose-release including steadily boosting the nutrient at the induction point to prolong cell growth and elevate cell densities greatly benefits the heterologous production of soluble RH in *E. coli*. As it has been demonstrated earlier that EnPresso-based results can be easily scaled to real glucose-limited fed-batch fermentations, the results provided should be an important basis for the design of a bioreactor-based process in larger volumes.

### 3.7. Biochemical and Spectroscopic Characterization of the Recombinant HoxBC and HoxC Proteins

The heterologously produced soluble HoxC_Strep_ and RH_stop_ proteins from TB batch and EnPresso cultures were analyzed by SDS-PAGE (Appendix A) and subsequently assayed for hydrogenase activity. However, no H_2_-mediated reduction of methylene blue was detected. The lack of activity prompted us to investigate the metal cofactor content in both proteins. Infrared (IR) spectroscopy was used to probe the ν(CO) (1870–2020/cm) and ν(CN) (2030–2150/cm) stretching modes of the CO and CN^−^ ligands of the [NiFe] catalytic center, which are observed in a spectral region devoid of protein contribution. Homologously produced HoxC exhibits two CO bands at 1941/cm and 1952/cm and broad CN^−^ signals at 2066/cm and 2084/cm, which are related to the two Ni_r_-S_I/II_ resting states of the catalytic center (Figure 7A) [14,18]. The IR spectrum of native RH_stop_, on the other hand, exhibits a dominant CO band at 1943/cm and two CN^−^ absorptions at 2070/cm and 2081/cm, which have been previously assigned to the catalytic Ni_a_-S intermediate (Figure 7B) [34]. The IR spectra of the heterologously produced HoxC and RH_stop_ proteins, however, revealed signals that are exclusively related to the protein backbone, i.e., amide II bands at 1548/cm. Thus, the IR data indicate that the lack of hydrogenase activity is related to the absence of the catalytic center. The lack of the catalytic center in the isolated HoxC protein was also supported by UV-visible spectroscopy, revealing the absence of characteristic [NiFe] site-related absorption bands previously observed in the homologously produced HoxC protein (Figure 7C) [14].

Although RH_stop_ isolated from *E. coli* showed neither detectable hydrogenase activity nor CO/CN-related bands in the IR spectrum, the protein preparations were characterized by a dark brownish color, suggesting the presence of iron-sulfur-clusters in the HoxB subunit. To gain insight into the FeS cluster composition, we performed EPR spectroscopy on as-isolated (oxidized) as well as reduced RH_stop_ samples. Reduction was accomplished through anaerobic incubation of RH_stop_ with an excess of sodium dithionite. While as-isolated RH_stop_ (Figure 7D, top) displayed only minor signals attributed to [3Fe-4S]^+^ clusters [34], the reduced sample (bottom) exhibited clear signals of [4Fe-4S]^+^ clusters (g_x_ = 2.043, g_y_ = 1.917, g_z_ =1.858), similar to those that were previously reported for the HoxB subunit isolated from *R. eutropha*. The EPR spectrum did not contain any signals related to the catalytic center, underling the fact that RH preparations originating from cultivation in *E. coli* lack the catalytic center.

In summary, the biochemical and spectroscopic characterization indicate the lack of the [NiFe] catalytic center in heterologously produced HoxC and RH_stop_. RH_stop_, however, seems to contain native-like iron-sulfur clusters in the HoxB subunit.

## 4. Discussion

In the present study, *R. eutropha* RH and its isolated HoxC and HoxB subunits were successfully produced in *E. coli* and purified from the soluble extract using a single-step affinity chromatography. In the case of RH overproduction, untagged HoxC was co-purified with Strep-tagged HoxB, indicating formation of the heterodimeric RH protein in *E. coli*, which is in good agreement with RH_stop_ forming a heterodimer in *R. eutropha* [33]. To improve RH production in *E. coli* BL21-Gold, experiments were conducted to identify optimal process parameters. Our results suggest that RH production is best upon induction of *hoxBC* gene expression with an IPTG concentration of 50 µM at a growth temperature of 30 °C. Lower culture temperatures decrease the risk of protein aggregation, because smaller protein production rates increase the likelihood of proper protein folding. In contrast, at a higher growth temperature such as 37 °C, recombinant proteins tend to form inclusion bodies [39,48,49]. Additionally, misfolded or aggregated proteins often induce the σ^H^-dependent stress response which, in turn, leads to an increased activity of heat-shock proteases and subsequently to an increased degradation of the aggregated proteins [50,51]. By contrast, the activity of heat shock proteases is reduced at lower temperatures resulting in a more stable protein production [52]. However, even after lowering the cultivation temperature to 20 °C, the most prominent contaminating band in our RH preparations was still present (Appendix A), indicating that a further decrease of the temperature would not improve the purity of the RH preparations.

Although the yield of overproduced RH_stop_ increased with increasing inducer concentration, our results showed that higher inducer concentrations negatively affect the solubility of the target protein. The use of higher inducer concentrations and the concomitant faster translation rates can exhaust the bacterial protein quality control systems and lead to improper folding of the newly translated proteins, resulting in inclusion bodies, as shown before in many studies (see e.g., p22 tailspike protein, amyloid, interleukin-1β) [53]. Consequently, IPTG-induced soluble RH production is most efficient at a low concentration of 20–50 µM in TB batch and fed-batch-like cultivation. Furthermore, we successfully applied the enzyme-based EnPresso system to produce difficult-to-express metalloproteins. Here, we achieved approximately 255 mg/L of purified RH, which is 18-times higher than that obtained in the initial batch cultures. Additionally, the production of RH in *E. coli* had a good scalability in shake-flask scales (250 mL- and 2.5 L-scale).

Surprisingly, no activity was detected in all RH_stop_ and HoxC samples purified from *E. coli,* although the proteins were soluble. This was due to the absence of the NiFe(CN)_2_(CO) cofactor, as confirmed by IR, EPR, and UV-vis spectroscopic data. However, these findings are not unexpected, as previous studies have demonstrated that active RH_stop_ can be heterologously produced in *E. coli* only under anaerobic cultivation conditions and upon co-production of the maturation proteins (HypA-F) from *R. eutropha* [19,20,21]. Moreover, expression of *E. coli hyp* genes and the specific nickel uptake system is controlled by the global transcriptional activator FNR (fumarate nitrate regulator) under anaerobic conditions [54,55]. Thus, the absence or a low level of active FNR under aerobic conditions [56] might contribute to the inactivity of the RH produced in aerobically grown *E. coli*. Furthermore, the absence of the [NiFe]-cofactor in the HoxC subunit is fully in line with the observed defect in hydrogen metabolism in the biotechnologically important model bacterium *E. coli* BL21 [57].

Nevertheless, our data are in support of the fact that the RH small subunit HoxB acquires its Fe–S clusters during synthesis in *E. coli* BL21. Interestingly, the mature HoxB subunit seems to form a complex with the [NiFe] cofactor-free HoxC subunit, revealing a partially mature RH. These data are quite surprising as it was previously believed that the RH possess an intrinsic proofreading mechanism preventing the complex formation of premature subunits [58]. The RH from *R. eutropha* belongs to the [NiFe]-hydrogenases lacking the C-terminal peptide extension of the large subunit. Most [NiFe]-hydrogenase large subunits carry such a C-terminal extension [1], which is cleaved off by an endopeptidase once the NiFe(CN)_2_(CO) catalytic center has been incorporated into the apo-protein [9]. The presence of this peptide sequence is supposed to prevent oligomerization of the small and large hydrogenase subunits [59,60,61]. However, a few [NiFe]-hydrogenases are naturally devoid of a C-terminal extension, yet they seem to employ the standard Hyp apparatus to synthesize the [NiFe] catalytic center. The RH from *R. eutropha* is one of them [23]. Both RH and the isolated catalytic HoxC are fully equipped with an intact NiFe(CN)_2_(CO) cofactor when purified from their original host *R. eutropha* [14]. Thus, a C-terminal extension is not required for either the subunit assembly or for the interaction with the Hyp apparatus [58]. Our data on the heterologously produced RH_stop_ suggest that HoxBC assembly is not triggered by the presence of the [NiFe] cofactor. Previous data have shown that the absence of the native Hyp proteins of *R. eutropha* results in a significantly lowered amount of both the HoxB and HoxC subunits in soluble *R. eutropha* extracts, suggesting a larger instability of at least the cofactor-free large subunit and thus preventing its proper interaction with the small RH subunit [62]. Surprisingly, this control mechanism seems to not function when the RH is overproduced in *E. coli* BL21. This obvious difference between the two hosts, *R. eutropha* and *E. coli*, was quite unexpected and deserves further investigation.

Taken together, compared to previous reports [19,20], our production system can provide excellent yields of soluble apo RH and HoxC proteins, suitable for structural and functional characterizations. The overall production yield obtained with our strategy exceeds the amount of RH gained in the purification of native RH from *R. eutropha* by several 100-fold [19,20]. Even though these proteins are only produced without [NiFe] cofactor, the availability of large quantities of recombinant precursors would allow to develop new artificial semisynthetic H_2_-producing catalysts in vitro, without the requirement of the canonical maturation machinery. This strategy has been proven successful in case of [FeFe]- and [Fe]-hydrogenases [22,63,64], where the incorporation of synthetic complexes contributed to the flourishing artificial enzymology.

## Figures and Tables

**Figure 1 microorganisms-09-01195-f001:**
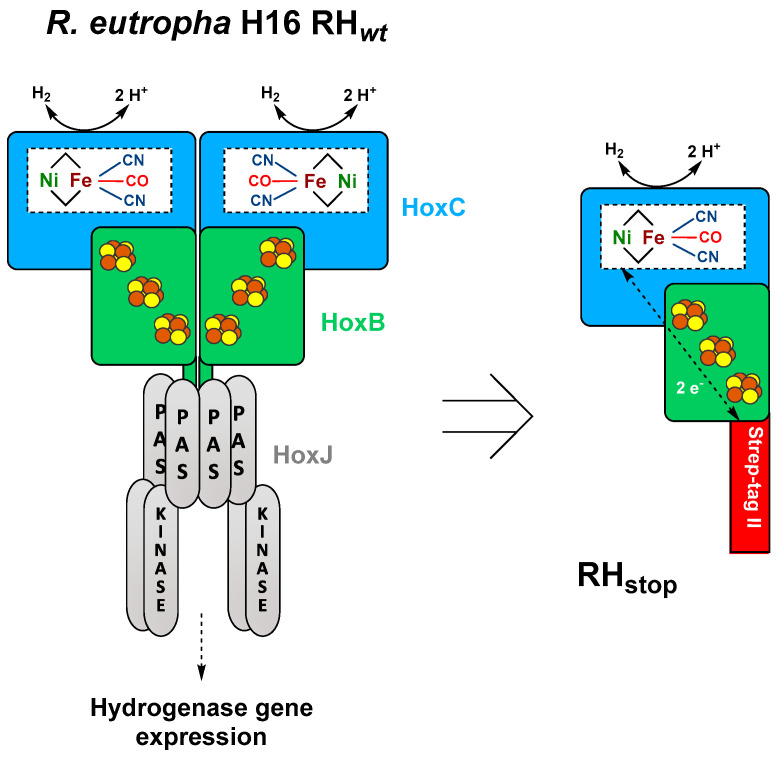
Schematic representation of the heterotetrameric native RH hydrogenase associated with the histidine kinase HoxJ (**left**) and the truncated heterodimeric RH_stop_ used in this study (**right**) adapted from the review of Lenz et al. [10]. The NiFe(CN)_2_(CO) active site is bound to the large subunit (blue) via four cysteine residues, while the small subunit (green) hosts three [4Fe–4S] clusters.

**Figure 2 microorganisms-09-01195-f002:**
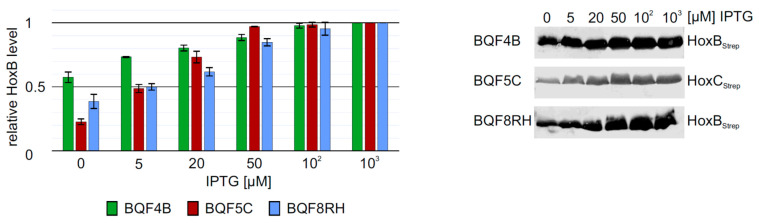
Heterologous overproduction of Hox proteins from plasmids pQF4, pQF5, or pQF8 in *E. coli* BL21-Gold (BQF4B, BQF5C, or BQF8RH). Cells were cultivated in TB medium in deep-well plates at 37 °C as described in Materials and Methods. RH production was induced with varying IPTG concentrations. HoxB production was analyzed by Western blotting (normalized to OD_600_) with Strep-tag antibodies (right part) and subsequent quantification of the bands using ImageJ. For each strain, protein amounts were calculated relative to the amount of protein at a maximum inducer concentration of 1 mM IPTG.

**Figure 3 microorganisms-09-01195-f003:**
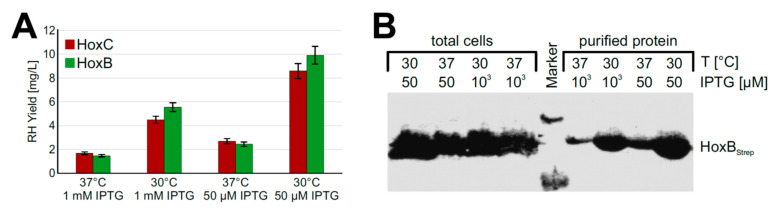
RH production in *E. coli* BQF8RH cultivated in 50 mL TB medium in UYF at 37 °C and 30 °C. RH production was induced with IPTG as indicated for 5 h. Soluble RH was purified by affinity chromatography and subsequently analyzed by SDS-PAGE and Western blotting. (**A**) Calculated amounts of purified RH after quantification of the stained gels in Appendix A with ImageJ; (**B**) Western blot of total protein and soluble protein with antibodies against the Strep-tag at HoxB.

**Figure 4 microorganisms-09-01195-f004:**
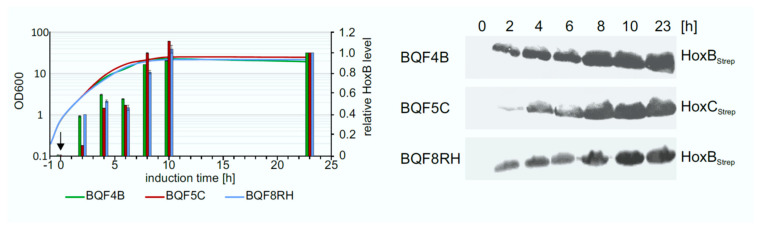
Time-dependent overproduction of HoxB_Strep_, HoxC_Strep_ and RH. Strains BQF4B, BQF5C, and BQF8RH were cultivated in 50 mL TB medium at 30 ^°^C in UYF. Samples normalized to OD_600_ of 5 were taken at the indicated time points after the addition of 50 µM IPTG (t = 0 h). Production of Hox proteins was analyzed by Western blotting. Growth curves of the different strains and relative Hox protein levels (**left**) as quantified from the Western blot (**right**). The individual protein levels after 23 h of induction were set to 1.

**Figure 5 microorganisms-09-01195-f005:**
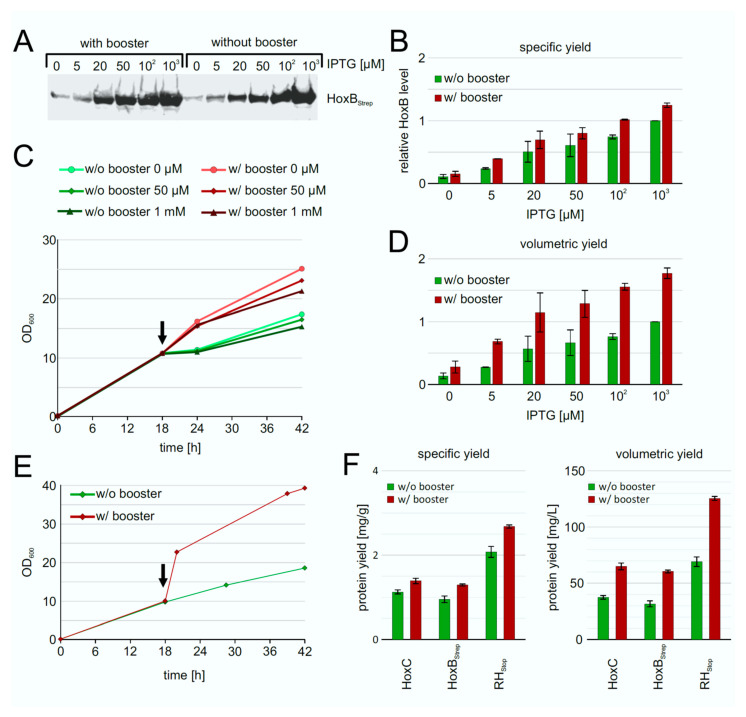
Heterologous production of RH by strain BQF8RH grown in EnPresso B medium with or without addition of booster. Cells were cultivated in deepwell plates at 30 °C as described in Materials and Methods. RH production was induced with varying IPTG concentrations. (**A**) Analysis of HoxB production by Western blotting with anti-Strep-tag antibodies. (**B**) Calculation of the specific HoxB yield normalized to the yield in non-boosted cultures induced with 1 mM IPTG. (**C**) Growth curve of representative cultures. The black arrow indicates the induction point. (**D**) Calculation of the volumetric HoxB yield normalized as in **B**. (**E**) Growth curves of strain BQF8RH in 50 mL EnPresso medium with or without booster in UYF. (**F**) Calculation of specific and volumetric RH yield from strain BQF8RH as grown in **E**.

**Figure 6 microorganisms-09-01195-f006:**
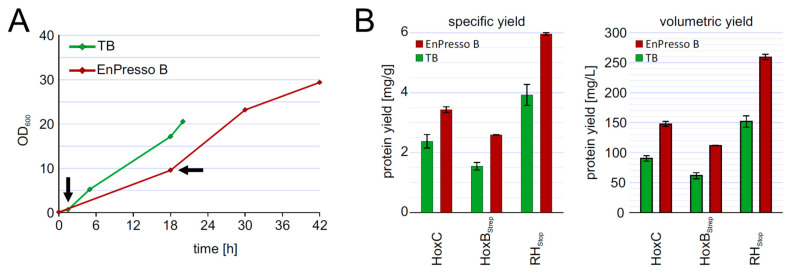
Up-scaling of RH production to 500 mL culture volume in 2.5 L UYF (**A**) growth curve of strain BQF8RH in complex TB or boosted EnPresso. (**B**) Calculation of specific and volumetric RH yield from strain BQF8RH as grown in **A**.

**Figure 7 microorganisms-09-01195-f007:**
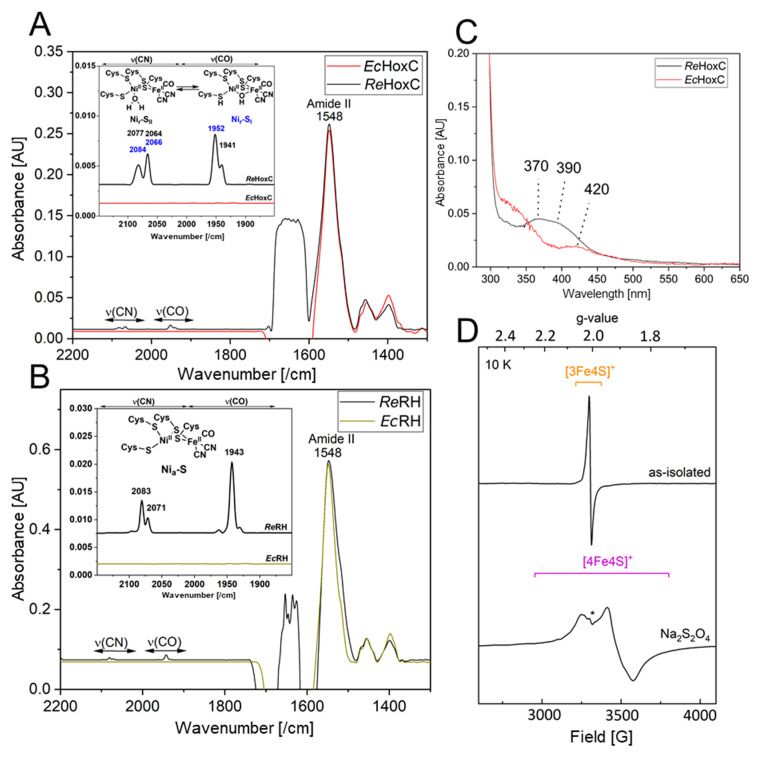
Comparative spectroscopic characterization of RH and isolated HoxC purified from *R. eutropha* and *E. coli*. (**A**) Comparison of IR spectra of as-isolated HoxC produced from *E. coli* (*Ec*HoxC, red line) and *R. eutropha* (*Re*HoxC, black line). (**B**) Comparison of IR spectra of as-isolated RH_stop_ produced from *E. coli* (dark yellow line) and *R. eutropha* (black line). (**C**) Comparison of UV-vis absorption spectra of 20 µM as-isolated HoxC produced from *E. coli* (red line) and *R. eutropha* (black line). The spectrum of *E. coli*-derived HoxC shows only minor absorptions around 420 nm, probably related to heme-containing protein contaminants copurified with the large subunit. HoxC from *R. eutropha* shows active site contributions characterized by two absorption bands at 370 and 390 nm [14]. (**D**) EPR spectra recorded at 10 K with a microwave power of 1 mW of as-isolated RH_stop_ (0.2 mM) from *E. coli* (top trace), containing minor contributions of a [3Fe–4S]^+^ cluster species, in line with data recorded of native RH [34]. The bottom trace represents reduced RH_stop_ treated with an excess of sodium dithionite. The spectrum contains rhombic signals attributed to a [4Fe–4S]^+^ cluster. The asterisk (*) denotes a weak signal, deriving presumably from a [2Fe–2S] cluster subspecies originating either from partial [4Fe–4S] cluster degradation or from protein contaminants. Spectra in **A**, **B** are normalized to the amide II band intensity at 1548/cm.

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
