# Peer review of "Optimization of Culture Conditions for Oxygen-Tolerant Regulatory [NiFe]-Hydrogenase Production from *Ralstonia eutropha* H16 in *Escherichia coli"

_microorganisms, 2021, doi:10.3390/microorganisms9061195_

Round 1

Reviewer 1 Report

I do not really see the great novelty nor impact in this study.

The EnBase/EnPresso system has been reported several times before

The effect of lower cultivation temp on the production of soluble protein in batch has long been known and has been reported in so many papers

The effect of the inducer concentration has to be explained in much more detail. is the T7 system tunable?

The product characterization is interesting, even though the product is catalytically inactive, but does not really fit to the rest of the manuscript - it seems like the authors tried to enlarge their study by adding this data

I miss a more elaborate investigation of recombinant protein production in the controlled bioreactor where growth rate and temperature can be decoupled. The study is limited by experiments done only in shake flasks and in obtaining inactive product.

Reviewer 2 Report

In this study, the authors report a heterologous expression procedure of the oxygen tolerant H2-sensing regulatory [NiFe]-hydrogenase (RH) from Ralstonia eutropha H16 in E. coli.

The authors improved the protein yield which is several 100-fold increase from that purified from the homologous host and made some important observations about the mechanism of subunit assembly although no activity was detected in the recombinant RH.

<Minor points>

Figure S2, upper panels

The value of molecular mass should be added at the bands in markers.

Line 225

“a second band was detected in the HoxB preparations”

The author should clearly indicate which is the second band.

Was the second band detected also in the sample derived from pQF8?

Figure 2, left panel, horizontal axis

IPTG [micromole] may be correct.

Line 271-272

“the overall productivity of soluble RH was nearly declined by half (Figure S3).”

This description does not agree with the values in Figure S3C.

Line 330 “booster”

Line 353 “by controlled limiting glucose supply to the cultures”

Readers who are not familiar with EnPresso may not be able to understand.

The author should explain the outline of EnPresso system.

//

Reviewer 3 Report

Fan et al. describe in their current manuscript the heterologous expression of an apoenzyme (= lacking the inorganic cofactor at the active site) of the regulatory hydrogenase of Ralstonia eutropha (now designated Cupriavidus necator) strain H16 in E. coli. The manuscript is overall well written and describes methods, results and outcome in an appropriate manner and adequate figures.

However, for such a relatively uninteresting outcome, I just think the of the manuscript is overly stretched and should be reduced in length. Maybe there are fans of heterologous expression and hydrogenase purification out there, but the average reader of the manuscript is most likely not interested in this sheer number of western blots, SDS-pages and bar graphs (most of which are recorded  of the enzyme under slightly refined expression conditions, when after all an inactive RH is what you get. I also hope the authors used replicates, this doesn't get clear, since no standard deviations are given/depicted (which should be the minimum in such bar graph figures, even better would be single dots depicted).

Figure 2 should be placed into the Supplement and I guess you mean µM.

Figure 3 One SDS-Gel as example and C) would be sufficient, the rest can be placed in Supplement.

Figure 4 can be placed in Supplement. After 2 hrs the order of colors is different (red in the middle) than after the rest (red is on the right), that is confusing.

I also think that the authors could tone down the "save the planet" (L11-12, L33-38) attitude in abstract and introduction, especially since it is not discussed afterwards. I am not surprised that this is not discussed, cause hydrogen production (or oxidation) is for sure not planned with an inactive apoenzyme of the probably least active hydrogenase out there.

Some line-for-line comments which should be reviewed:

L11-12: Why not simply delete these lines and start with the fact that most hydrogenase enzymes are difficult to produce?

L17: I would argue against that the RH is simple, even not relatively simple.

L18: you could include the strain here. In turn you could delete the 24 deepwell plates (if at all, you could give a volume here).

L19: Give a result here: What were the most relevant cultivation parameters? Delete initially.

L20: Delete significantly

L21: Maybe you can rephrase the monster "EnPresso B-based fed-batch-like cultivation". What are "shake flasks"? Are they much different from simple "flasks" (maybe Erlenmeyer)? Is it important in the abstract?

L22: hundredfold. Delete additionally or also.

L33-36: As above you could simply delete these lines without losing much information.

L38: Ref 2 is not really appropriate as there are much better reviews more specific for this. To be honest this reads like an inappropriate self-citation. If you want to cite that review, place it elsewhere and search for a better citation.

L39: Here again, Ref 5 suits better and is newer.

L43: Again, Ref 2 is not really the best source, see above!

L44: I really question that the authors really read their references. Here, only 6 and 7 are appropriate. What the hell are antibiotic resistance genes (ref 8) have to do with hydrogenases? Ref 9 is also not really great here.

L63-64: Could you assign the references to the statement (i.e. which ref belongs to heterologous and which to homologous expression)?

L65: It is not simple.The overall structure is similar to most other NiFe hydrogenases studied. It is also not really a model system just because you can cite two articles, of which one is a review (I would delete ref 11 here). For most studies on the catalytic center, other NiFe hydrogenases were chosen. 

Figure 1 might lead the reader to the wrong conclusion - that you can produce an enzyme with an active site. Either redo the figure and/or make it clear in the legend.

L76 - 84 are a bit too detailed, shorten. Use the free space to write a few lines about the maturation of hydrogenases, since the reader might need that information for the discussion and some parts of the results.

L90: No need to cite sambrook for TB medium! You might state a company if you bought TB though.

L95/98: Delete "annealed" (self-explanatory what the oligos do here)

L108: Delete comma

L116: volume of the wells?

L119-121: How were the cells cultivated before the deepwell plate? Did you shake the plates?

L119/137: I miss details about the OD measurement, which cuvettes and/or which apparatus was used, when was diluted, etc.

L124: Biocatalyst dose?

L126: Which Shaker, what rpm?

L133: how much shaking?

L139: comma before booster

L152: sonication

L157: It doesn't get clear where you performed filtration.

L202-207 is introduction and part of it redundant. Delete/place in introduction.

L207-216 is more methods and partly described, partly self-explanatory (strep-tag allows affinity chromatography? Really?). Delete most of it.

L217-219: As you said, already described. Also, L217 contains an error.

Shorten L220-229 you might also combine with the following chapter.

L220: standard sizes are not given for S2,

L226: Size of smaller band should be mentioned. I was looking for a band at a really smaller size until I noticed that it is directly below HoxB.

L231-245 can also be shortened (often, methods are repeated or sentences can be streamlined to give the same information).

L254-260: Most of this can be deleted.

L261: I was surprised by the many bands you have in the SDS-Page, this should be briefly presented.

L271: While not quantified, it seems that the "contaminant" bands are lower at 20°C. And/or the overall protein amount seems less.

L271: Productivity of soluble RH? I thought the enzyme is not active... was declined is also incorrect.

L280: To be honest, the blots and gels often don't look like a good quantification is possible. How did you check for saturation and similar? In my experience this is difficult to handle.

L284-296: This are minor results and this part can be shortened.

L298-303 can be deleted.

L297-313 also minor results, which can be shortened/supplement.

L322-328 is introduction.

L349: was

L366: write out the KLa value here or in methods.

L389-390: Which samples did you use for activity? All of them? How much and how long did you wait?

L393-398: Shorten and give this information after your result.

L399: It doesn't get clear which sample you used for spectroscopy. I am bothered that the signals could be attributed to some other bands - depending on the purification there are some other bands clearly visible on the gel. Were any attempts to further purify undertaken? If not, critically discuss.

L451: Some words on the purification, the impurities detected there, how to improve that would be good.

L452-466 can be compressed.

L472: Maturases? Also, is it really necessary to state lines 473-476 and cite refs 56-58? This makes the impression that BL21 could 1) not produce any nickel enzymes, which is not true and 2) produce NiFe hydrogenases if just this is fine. I think you can't make any statements if these three defects would really decrease NiFe hydrogenase production if the "maturases" are present.

L483-496: It could still be that there is a proofreading process (and other described processes) in Ralstonia/Cupriavidus. Please don't overinterpret your results in a heterologous host! Shorten/delete this.

L499-501: Is it instable or is "just" less produced? Also, proofreading mechanism is too much interpretation in my opinion (maybe it is just not as much produced - or it is instable, which is not really a proofreading mechanism).

L503: What is an excellent yield? You describe later more appropriate by comparing to previous results.

L507: What exactly is a significant quantity?

Author contributions: I don't know how strict microorganisms/MDPI is with their own rules, but according to the authorship instructions, authors should be only people who

  • Substantial contributions to the conception or design of the work; or the acquisition, analysis, or interpretation of data for the work; AND
  • Drafting the work or revising it critically for important intellectual content; AND
  • Final approval of the version to be published; AND

https://www.mdpi.com/journal/microorganisms/instructions#authorship 

Corresponding to this, PN and OL do not qualify as authors, but should be listed in the acknowledgement. You might discuss this with the editor.

Round 2

Reviewer 1 Report

I want to thank the authors for their answers. However, I still do not see the novelty nor the impact in this study.